

# Gender-related disparities in the frequencies of PD-1 and PD-L1 positive peripheral blood T and B lymphocytes in patients with alcohol-related liver disease: a single center pilot study

Beata Kasztelan-Szczerbinska[1], Katarzyna Adamczyk[1], Agata Surdacka[2], Jacek Rolinski[2], Agata Michalak[1], Agnieszka Bojarska-Junak[2], Mariusz Szczerbinski[3] and Halina Cichoz-Lach[1]

[1] Department of Gastroenterology with Endoscopy Unit, Medical University of Lublin, Poland, Lublin, Poland
[2] Department of Clinical Immunology, Medical University of Lublin, Poland, Lublin, Poland
[3] Department of Gastroenterology with Endoscopy Unit, Public, Academic Hospital No 4, Lublin, Poland

Corresponding author
Beata Kasztelan-Szczerbinska,
beata.szczerbinska@op.pl

## ABSTRACT

**Background**. Exposure to excessive alcohol consumption dysregulates immune signaling. The programed cell death 1 (PD-1) receptor and its ligand PD-L1 play a critical role in the protection against immune-mediated tissue damage. The aim of our study was evaluation of the PD-1/PDL-1 expression on peripheral T and B lymphocytes, its correlation with markers of inflammation and the severity of liver dysfunction in the course of alcohol-related liver disease (ALD).

**Material and Methods**. Fifty-six inpatients with ALD (38 males, 18 females, aged 49.23 ± 10.66) were prospectively enrolled and assigned to subgroups based on their: (1) gender, (2) severity of liver dysfunction (Child-Pugh, MELD scores, mDF), (3) presence of ALD complications, and followed for 30 days. Twenty-five age- and gender-matched healthy volunteers served as the control group. Flow cytometric analysis of the PD-1/PD-L1 expression on peripheral lymphocyte subsets were performed.

**Results**. General frequencies of PD-1/PD-L1 positive T and B subsets did not differ between the ALD and control group. When patients were analyzed based on their gender, significantly higher frequencies of PD1/PD-L1 positive B cells in ALD females compared to controls were observed. ALD females presented with significantly higher frequencies of PD-1+ and PD-L1+ B cells, as well as PD-L1+ all T cell subsets in comparison with ALD males. The same gender pattern of the PD-1/PDL1 expression was found in the subgroups with mDF > 32 and MELD > 20. No correlations of PD-1+ and PD-L1+ lymphocyte percentages with mDF, CTP and MELD scores, nor with complications of ALD were observed. Significant correlations of PD-L1 positive B cell frequencies with conventional markers of inflammation were found.

**Conclusions**. Gender-related differences in the frequencies of PD-1/PD-L1 positive T and B cells were observed in patients with ALD. Upregulation of PD-1+/PD-L1+ lymphocytes paralleled both the severity of alcoholic hepatitis and liver dysfunction in ALD females.

# INTRODUCTION

Alcohol abuse remains a huge problem in the Western world. The available body of evidence indicates that it is associated with the increased prevalence of different chronic disorders i.e., cancers, lung and cardiovascular diseases as well as liver cirrhosis (*Yoon & Chen, 2018*, https://pubs.niaaa.nih.gov/publications/surveillance111/Cirr15.pdf). According to data from the National Institute on Alcohol Abuse and Alcoholism, over 80,000 deaths per year in the United States are attributed to alcohol misuse (*National Instituteon Alcohol Abuse and Alcoholism, 2017*, https://www.niaaa.nih.gov/alcohol-health/overview-alcohol-consumption/alcohol-facts-and-statistics). Also, the high mortality rate due to liver cirrhosis in Poland and other Central and Eastern European countries, as well as in the United Kingdom, Ireland, and Finland (*World Health Organization (2014)*, http://www.who.int/healthinfo/statistics/mortality_rawdata/en/; *Bosetti et al., 2007*) is related mainly to the high per capita alcohol consumption.

There is a sufficient body of evidence, that alcohol-attributable end-stage liver disease and liver cancer are entirely preventable (*Sheron, 2016*; *Rehm, Samokhvalov & Shield, 2013*). Therefore, it is vitally important for policy planning to chart the methods leading to the reduction of harmful alcohol consumption. Furthermore, also systematic research in order to implement new diagnostic and therapeutic tools and reach new conclusions should be foster. In comparison to huge advances made in the management of viral hepatitis (vaccines and oral therapies for HBV, oral regimes for HCV), alcohol-related liver disease (ALD) management has lagged. Moreover, patients with ALD are generally identified at the late stages of the disease, and programs for early detection are scarce (*Ndugga et al., 2017*).

Exposure to chronic and excessive alcohol consumption, its breakdown metabolites and gut-derived endotoxins dysregulate immune signaling and give rise to activation of the local and systemic pathways of inflammation (*Gao & Tsukamoto, 2016*; *Szabo & Saha, 2015*). Toxic ethanol metabolites including reactive oxygen species (ROS) activate T and B cell clones against self- and modified proteins. Activated B cells can produce immunoglobulins directed both to haptens and native antigens (*Szabo & Saha, 2015*; *Molina et al., 2010*). As a result, the non-resolving inflammatory response may occur leading to ALD development and progression. Furthermore, ethanol misuse contributes to the release of circulating modulators of immunity and inflammation, that affect multiple organs and tissues, potentially causing their failure (*Louvet & Mathurin, 2015*; *Hernaez et al., 2017*). Therefore, the aforementioned responses need to be properly controlled to maintain the mechanisms of tolerance and immune homeostasis.

Recent studies have elucidated the relevance of the programmed cell death 1 (PD-1) receptor and its ligand PD-L1 in inhibition of self-reactive and effector cells and the protection against immune-mediated tissue damage. As a negative regulator, PD-1 exerts

a suppressive effect on previously activated T and B cells by binding to its ligands PD-L1/PD-L2. It results in inhibition of antigen-specific immune cell proliferation, cytokine release, and cytolytic function. PD-L1 is also constitutively expressed on T and B cells, dendritic cells (DCs), macrophages, mesenchymal stem cells and bone marrow-derived mast cells (*Yamazaki et al., 2002*; *Boussiotis, Chatterjee & Li, 2014*). PD-1 conducts signals only when it is cross-linked with B- or T-cell antigen receptors. These coinhibitory signals modulate the intensity and duration time of immune reactions and may restrict immune-induced tissue damage, control the resolution of inflammation, and maintain peripheral immune tolerance (*Bardhan, Anagnostou & Boussiotis, 2016*; *Saresella et al., 2012*).

The PD-1/PD-L1 pathway has been shown to play an important role in a variety of diseases, like cancer, autoimmune conditions, and chronic infections. It has already been investigated using experimental animal models in systemic lupus erythematosus, encephalomyelitis, myasthenia gravis, diabetes mellitus, myocarditis, inflammatory bowel diseases, systemic sclerosis, and rheumatoid arthritis. The critical role of PD-1/PD-L1 signaling in the prevention of disorders caused by impaired immune activation has become evident (*Dinesh, Hahn & Singh, 2010*; *Zamani et al., 2016*). However, its impact on the development and progression of ALD has not been entirely explained yet. It is likely that alterations of the checkpoint inhibitor surveillance may be responsible for long-lasting activation of the inflammatory response in the liver, as well as the systemic character of the disease. Recently *Markwick et al. (2015)* reported that PD-1, and T-cell immunoglobulin and mucin domain-containing protein 3 (TIM3), as well as their ligands i.e., PD-L1 and galectin-9, represented relevant elements of innate and adaptive immunity in the course of alcoholic hepatitis (AH). T lymphocytes from AH patients showed higher expression of PD-1 and PD-L1 in comparison with control T cells. They indicated that PD-1 and TIM3 blockade might create the potential new approach for the treatment of the disease.

On this background, we aimed to explore possible alterations in the expression of PD-1/ PD-L1 proteins on peripheral T and B lymphocytes, their eventual correlations with conventional markers of inflammation and the severity of alcoholic hepatitis (AH), as well as with the severity of liver dysfunction classified according to the criteria (Child-Turcotte-Pugh, Model of End-Stage Liver Disease, modified Maddrey's discriminant function) widely used for this purpose in clinical settings for ALD patients.

Since a great body of evidence indicates that compared with their male counterparts, women are more susceptible to the toxic effects of ethanol in the liver for any given dose of alcohol (*Chou, 1994*; *Greenfield, 2002*; *Grant et al., 2017*; *Lowe et al., 2019*), and also symptoms and signs in ALD demonstrate sexual dimorphisms, possible gender-related differences in the PD-1/PD-L protein expression were also assessed. Sex hormones have been suggested to influence immune response. These hormones exert their biological effects by binding to the inherent receptors on immune cells, what results in further modification of gene expression, lymphocyte proliferation, antigen presentation, and cytokine secretion (*Whitacre, 2001*). Females are known to produce higher titers of circulating immunoglobulins, as well as a major variety of autoreactive antibodies. Women also demonstrate a more pronounced humoral immune response against infections in comparison with men (*Arroyo & Montor, 2011*). Understanding the background of

gender-related differences in the ALD course may be relevant for individually tailored therapies.

## MATERIALS & METHODS

### Characteristics of the studied cohort

We used the same patient recruitment protocol as presented in our previous studies (*Kasztelan-Szczerbinska et al., 2014*; *Kasztelan-Szczerbińska et al., 2015*). In short, 56 consecutive inpatients (18 females and 38 males) with ALD were prospectively enrolled over 2 years. Twenty- five healthy volunteers who matched for age and sex (nine women and 16 men) served as the control group. Controls confirmed abstinence or declared that their alcohol consumption had not exceeded 10 g of ethanol daily. The diagnosis of alcohol-related liver disease was based on classic symptoms and signs revealed after a patient's history taking and physical examination in combination with laboratory alterations i.e., increased liver enzyme levels and AST / ALT ratio greater than 2 typically for ALD, as well as results of imaging studies in individuals with alcohol misuse. Taking into account the disease indicators well described and calibrated in prior clinical research, the identification and characterization of the illness in our studied cohort was based on the recommendations of EASL and AASLD without performing a liver biopsy (*Stickel et al., 2017*). Other etiology of chronic liver injury were ruled out. Alcohol drinking habits were established using the Alcohol Use Disorder Identification Test (AUDIT). A score of less than 5 was serving as a signal of a non-drinker, while an AUDIT score of 8 or more was an indication of harmful drinking. The average drinker score was 32 (*Saunders et al., 1993*). The positive AUDIT result served as the study inclusion criterion.

Heavy and hazardous alcohol consumption in studied patients was defined by The World Health Organization criteria (http://www.who.int/gho/alcohol/consumption_patterns/heavy_episodic_drinkers_text/en/). In our cohort, the female daily etanol intake ranged from 40 g/d to more than 100 g/d, and male daily intake from 60 g/d to more than 100 g/d. No treatment was administered to any patient at the enrollment. Patient demographics was collected and the established procedures were performed and finished for each individual within 48 h at hospital admission. The Child-Turcotte-Pugh (CTP) (*Peng, Qi & Guo, 2016*), MELD (Model End-Stage Liver Disease) (*Ashwani, Singal & Kamath, 2013*) and modified Maddrey Discriminant Function (mDF) score were used to categorize the liver dysfunction (internet calculators were applied from https://www.mdcalc.com were used).

Studied patients were distributed to subgroups based on their:
1. gender,
2. degree of liver failure classified by CTP, MELD and mDF scores
3. simultaneous medical conditions related to ALD decompensation (ascites, hepatic encephalopathy- HE, esophageal varices, kidney dysfunction).

Other potential confounders as blood transfusions during last 6 months before the study entry, history of immune-related comorbidities or allergies, and immunomodulators administration were excluded. Additional severe coexisting disorders such as malignancy,

respiratory failure, severe cardiovascular dysfunction, or unstable diabetes mellitus, were also ruled out.

West-Haven criteria (*Dharel & Bajaj, 2015*) were applied to classify signs of overt hepatic encephalopathy. The presence of ascites was checked by ultrasound examination. Esophageal varices were identified by endoscopic examination of upper gastrointestinal tract. Kidney dysfunction was defined by blood creatinine values above 1.2 mg/dl (i.e., the upper limit of normal).

Studied patients were discharged home after liver function parameters stabilized, as well as physical and emotional symptoms and signs of alcohol withdrawal resolved. Follow-up outpatient visits were scheduled every 2 weeks during one month after patient discharge or during every hospital admission if needed.

Five (8.9%) patients died within 30 days of follow-up.

## Ethical Requirements

All individuals signed the written informed consent prior to their inclusion in the study with respect to examining their blood samples for scientific purposes and were free to withdraw at any time without providing a reason. Strict confidentiality was maintained throughout the process of data collection and analysis. The study protocol conforms to the ethical guidelines of the 1975 Declaration of Helsinki (6th revision, 2008) as reflected in a priori approval by the institutional review board of Medical University of Lublin (KE-0254/141/2010).

## Laboratory examinations

Analyses of basic laboratory tests included as follows: liver enzymes (transaminases- ALT, AST, alkaline phosphatase-AP, gamma-glutamyl transpeptidase- GTP), liver function parameters (total bilirubin, albumin, urea, prothrombin time and INR), complete blood count, parameters of renal function, and conventional markers of inflammation (white blood cells count, neutrophils count, neutrophil to lymphocytes ratio, C-reactive protein level).

## Cell isolation and flow cytometric analysis

Immunofluorescence examinations were performed based on the combination of allophycocyanin (APC), phycoerythrin (PE)- and fluorescein isothiocyanate (FITC)-labeled monoclonal antibodies (mAbs). All mAbs (BD Biosciences, USA) applied in our study are presented in the Table S7. Five milliliters of the whole blood obtained by venipuncture from ALD patients and healthy controls were collected into sterile, lithium heparin-treated blood collection systems (S-Monovette, SARSTEDT AG & Co., D-51588 Numbrecht, Germany) and incubated with mAbs (20 μl per test) for 20 min. in the dark at room temperature. Next, BD FACS™ Lysing Solution (BD Biosciences, USA) was applied for 10 min in the dark to lyse red blood cells. Subsequently, the samples were washed with Phosphate Buffered Saline (PBS) (Sigma-Aldrich, Germany) solution and analyzed by flow cytometry directly after preparation. A *FACSCalibur™ flow cytometer* (BD Biosciences) equipped with 488 nm argon laser was used for data acquisition and analysis. A minimum of 10,000 events for each analysis were acquired and analyzed with CellQuest Software. For

each person, lymphocytes were identified and gated by setting appropriate forward and side scatter parameters. Examples of the cytometric analysis are presented in Figs. S4 and S5.

## Statistical analysis

Statistical analysis was performed using the Statistica 10 software package (StatSoft, Poland). The distribution of the data in the groups was preliminarily evaluated by Kolmogorov and Smirnov test. A skewed distribution of checked values was found, so continuous variables were presented as medians with interquartile range and assessed using the Mann–Whitney U test. Categorical variables were described as numbers with percentages and compared using either Fisher's exact test or the $\chi 2$ test as appropriate. The differences in the frequencies of PD-1 and PD-L1 positive lymphocytes between CTP classes were analyzed using Kruskal-Wallis and posthoc Dunn's multiple comparisons test. Spearman's rank correlation test was used for the assessment of correlations between the frequencies of PD-1/PD-L1 positive T and B cells and liver function parameters, as well as standard indices of inflammation. The receiver operating curves (ROC) were constructed and their areas under the curve (AUCs) checked in order to estimate potential accuracy of selected frequencies of PD-1/PD-L1 positive lymphocytes as a discrimination measure in liver dysfunction modelling. The method of *DeLong, DeLong & Clarke-Pearson (1988)*, for the calculation of the Standard Error of the AUC was used. Youden's $J$ statistic was used to define the optimized cut-off values (*Schisterman et al., 2005*). A two-sided *p*-value of less than 0.05 was considered to be associated with statistical significance.

## RESULTS

### Basic characteristics of the studied groups

Fifty-six patients (pts) met the inclusion criteria and were enrolled in the study, including 38 males (67.9%) and 18 females (32.1%). Their mean age was $49.54 \pm 10.94$ and $47.78 \pm 12.22$, respectively. Five (8.92%) of 56 pts with ALD died from complications of liver failure within 30 days of follow up. The matching control group consisted of 25 healthy volunteers who consumed no more than 10 g of ethanol per day including 16 (64.0%) males and nine (36.0%) females aged $46.21 \pm 11.23$ and $45.11 \pm 10.23$, respectively. Several surveys have indicated that the relative risk of alcohol-associated liver injury is higher in females in comparison to males (*Loft, Olesen & Døssing, 1987*; *Becker et al., 1996*; *Rehm et al., 2007*). Therefore, patients were assigned to two subgroups based on their gender. Nevertheless, both subgroups have not differed significantly regarding their liver function parameters nor demographic data. The baseline characteristics of ALD patients and the control group is summarized in Table 1.

### Analysis of the PD-1 and PD-L1 positive T and B lymphocyte frequencies in the peripheral blood of patients with ALD in comparison with the control group and gender-related differences

The general frequencies of PD-1 and PD-L1 positive T and B subsets did not differ between the ALD and the control group (see Table S1). Although, when the ALD group were

**Table 1  Demografic and laboratory data of the studied group.**

| Variable | ALD group | | | | p | Controls | | | | p |
|---|---|---|---|---|---|---|---|---|---|---|
| | Males (n = 38) | | Females (n = 18) | | | Males (n = 16) | | Females (n = 9) | | |
| | Median | 5–95 percentiles | Median | 5–95 percentiles | | Median | 5–95 percentiles | Median | 5–95 percentiles | |
| Age/years/ | 49.00 | 33.00–64.00 | 51.50 | 25.00–67,00 | 0.62 | 42,50 | 29,00–53,00 | 37.00 | 28.00–72.00 | 0.65 |
| ALT IU/L | 48.50 | 19.40–250.00 | 39.50 | 21.00–240.00 | 0.33 | 27.50 | 21.30–30.70 | 28.00 | 22.00–30.00 | 0.80 |
| AST IU/L | 111.00 | 41.85–503.45 | 102.50 | 39.00–261.00 | 0.40 | 24.50 | 20.60–27.40 | 26.00 | 22.00–28.00 | 0.26 |
| ALP IU/L | 140.00 | 75.00–492.15 | 136.00 | 71.05–348.70 | 0.80 | 91.50 | 73.60–102.00 | 98.00 | 77.00–111.00 | 0.26 |
| GTP IU/L | 390.00 | 60.80–2394.15 | 410.50 | 92.60–521.80 | 0.39 | 10.50 | 9.00–12.70 | 10.00 | 9.00–12.00 | 0.72 |
| Bil mg/dL | 3.40 | 1.00–19.50 | 2.05 | 0.60–28.10 | 0.69 | 0.80 | 0.50–1.00 | 0.70 | 0.30–0.90 | 0.17 |
| Alb g/dL | 2.96 | 1.93–4.02 | 3.28 | 2.25–4.64 | 0.19 | 4.07 | 3.87–4.23 | 4.11 | 3.79–4.23 | 0.93 |
| INR | 1.41 | 0.93–2.67 | 1.43 | 0.75–2.44 | 0.71 | 0.90 | 0.90–1.10 | 1.00 | 0.90–1.10 | 0.66 |
| Crea mg/dL | 0.70 | 0.44–1.73 | 0.60 | 0.44–2.22 | 0.26 | 0.80 | 0.60–1.07 | 0.75 | 0.60–1.00 | 0.55 |
| CRP mg/L | 21.71 | 1.75–81.75 | 26.40 | 0.57–122.58 | 0.73 | 1.84 | 0.65–3.41 | 1.97 | 0.98–3.12 | 0.71 |
| WBC × $10^3$ cells/μL | 6.88 | 3.43–17.16 | 7.905 | 3.262–23.580 | 0.69 | 5.15 | 4.30–6.88 | 5.46 | 4.51–6.21 | 0.59 |
| NEUT × $10^3$ cells/μL | 4.59 | 2.12–15.23 | 4.88 | 1.52–20.28 | 0.74 | 3.18 | 2.78–4.36 | 3.41 | 2.68–3.80 | 0.63 |
| NLR | 4.33 | 1.64–14.62 | 3.64 | 1.45–22.08 | 0.51 | 2.21 | 1.80–2.51 | 1.98 | 1.61–2.30 | 0.43 |
| PLT × $10^3$ cells/μL | 128.00 | 46.15–236.15 | 155.00 | 54.00–412.20 | 0.09 | 312.50 | 258.00–354.60 | 291.00 | 233.00- 333.00 | 0.09 |
| RBC × $10^6$ cells/μL | 3.47 | 2.53–4.75 | 3.80 | 2.73–248.69 | 0.24 | 5.13 | 4.81–5.39 | 4.23 | 4.11 –5.00 | <0.001 |
| HGB g/dL | 11.50 | 8.53–15.27 | 12.25 | 8.94–16.86 | 0.17 | 15.80 | 14.83–16.27 | 12.90 | 12.30-13.10 | <0.001 |
| MELD | 16.00 | 6.45–30.55 | 14.00 | 6.40–34.60 | 0.39 | | | | | |
| mDF | 18.94 | 0.91–97.706 | 15.94 | 0.29–100.02 | 0.48 | | | | | |
| CTP class | | | | | | | | | | |
| A | 9 (23.68%) | | 5 (27.78%) | | 0.66 | | | | | |
| B | 16 (42.10%) | | 9 (50.00%) | | | | | | | |
| C | 13 (34.21%) | | 4 (22.22%) | | | | | | | |
| ALD complications | | | | | | | | | | |
| Ascites | 18 (47.37%) | | 9 (50.00%) | | 0.79 | | | | | |
| HE | 5 (13.16%) | | 1 (5.55%) | | 0.41 | | | | | |
| EV | 20 (52.63%) | | 9 (50.00%) | | 0.98 | | | | | |
| Non-survival | 4 (10.53%) | | 1 (5.55%) | | 0.56 | | | | | |

**Notes.**

Alb, albumin (normal range [NR] 3.2–4.8); ALD, alcohol-related liver disease; ALT, alanine aminotransferase (NR < 31); ALP, alkaline phosphatase (NR 45–129); AST, aspartate aminotransferase (NR < 34); Bil, bilirubin (NR 0.3–1.2); Crea, creatinine (NR 0.5–1.1); CRP, Creactive protein (NR 0.0–5.0); CTP, Child-Turcotte- Pugh score; EV, esophageal varices; GGT, gamma-glutamyl transpeptidase (NR < 50.0); HE, hepatic encephalopathy; Hgb, hemoglobin (NR 14.0–18.0); INR, International Normalized Ratio (NR 0.8–1.2); mDF, the Maddrey's Discriminant Function; MELD, Model for End-Stage Liver Disease; NEUT, neutrophils (NR 1.8–7.7); NLR, neutrophil to lymphocyte ratio; p, significance level; PLT, platelets (NR 130400); RBC, red blood cells (NR 4.5–6.1); WBC, white blood cells (NR 4.8–10.8).

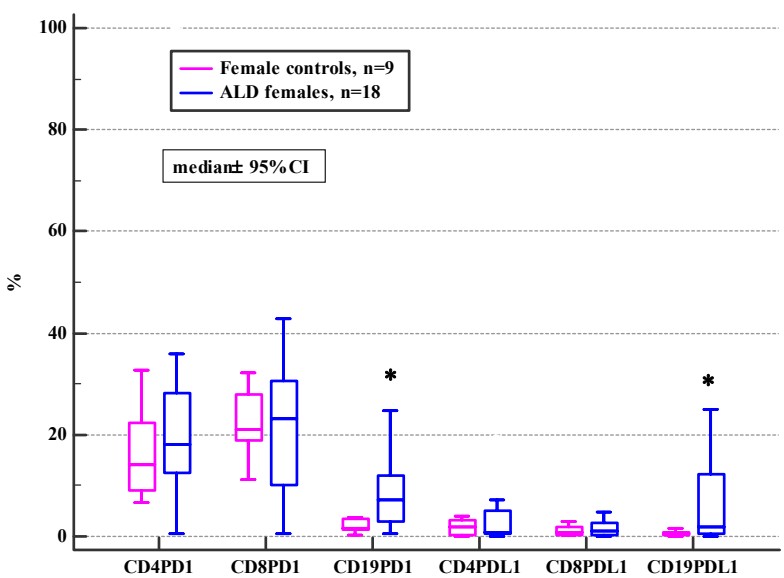

**Figure 1** Comparison of the frequencies (%) of PD-1 and PD-L1 positive T and B cells in ALD females versus female controls (* $p < 0.05$).

analyzed based on their gender, significantly higher frequencies of PD1 and PD-L1 positive CD19+ B cells in ALD females compared to female controls were observed (Fig. 1; Table 2).

There were no statistically significant differences in the frequencies of PD-1/PD-L1 positive T and B cells between the ALD and control groups of the studied men (Table 2).

Furthermore, females with ALD presented with significantly higher frequencies of PD-1 and PD-L1 positive B cells, as well as PD-L1 positive T cells (both subsets) in comparison with ALD males (Table 2).

There were no gender-related differences in the frequencies of PD-1/PD-L1 positive T and B lymphocytes in the control group (Table 2).

## Analysis of mutual correlations between frequencies of PD-1/PD-L1 positive B and T lymphocyte subsets in patients with ALD

Significant correlations were observed between frequencies of PD-1 positive CD4+ and CD8+ cells (Rho 0.75), as well as PD-L1 positive CD4+ and CD8+ cells (Rho 0,86). Furthermore, significant correlations were confirmed between frequencies of PD-1 positive B cells (CD19+) and both PD-L1 positive T subsets (Rho 0.37 for CD4+; Rho 0.41 for CD8+), as well as PD-L1 positive B (Rho 0.39) and PD-L1 positive CD4+ (Rho 0.33) and CD8+ (Rho 0.35) T cells. Furthermore, frequencies of PD-L1 positive B cells (CD19+) showed a positive correlation with both PD-L1 positive T cell subsets (Rho 0.75 for CD4+; Rho 0.77 for CD8+). (Rho= Spearman's correlation coefficient) (Table S8). Similar mutual correlations were seen for CD19+ subsets in the control group (Table S7).

Kasztelan-Szczerbinska et al. (2021), *PeerJ*, DOI 10.7717/peerj.10518

**Table 2** Comparison of the frequencies (%) of PD-1/PD-L1 positive T and B lymphocytes inALD and controls based on patients' gender (a Mann-Whitney test).

| Variable | ALD group | | | | p1* | Controls | | | | p2* | p3* | p4* |
|---|---|---|---|---|---|---|---|---|---|---|---|---|
| | Males ( *n* = 38) | | Females ( *n* = 18) | | | Males ( *n* = 16) | | Females ( *n* = 9) | | | | |
| | Median | 5–95 percentile | Median | 5–95 Percentile | | Median | 5–95 Percentile | Median | 5–95 Percentile | | | |
| **PD-1 positive cell subsets** | | | | | | | | | | | | |
| CD4+ | 18.44 | 1.83–3.28 | 18.07 | 1.43–4.03 | 0.73 | 21.07 | 8.40–9.64 | 14.20 | 6.55–30.67 | 0.17 | 0.18 | 0.60 |
| CD8+ | 19.40 | 2.68–6.57 | 23.06 | 0.67–6.70 | 0.99 | 25.57 | 7.55–7.51 | 21.11 | 11.28–43.08 | 0.68 | 0.22 | 0.63 |
| CD19+ | 3.27 | 0.36–4.20 | 7.08 | 0.69–0.76 | 0.02 | 2.54 | 0.78–9.39 | 1.66 | 0.35–14.70 | 0.47 | 0.59 | 0.04 |
| **PD-L1 positive cell subsets** | | | | | | | | | | | | |
| CD4+ | 0.65 | 0.04–2.76 | 0.90 | 0.09–8.04 | 0.04 | 1.06 | 0.26–5.10 | 1.83 | 0.10–4.03 | 0.80 | 0.09 | 0.44 |
| CD8+ | 0.44 | 0.03–1.66 | 1.14 | 0.02–7.21 | 0.02 | 0.97 | 0.13–2.43 | 0.89 | 0.18–2.88 | 1.00 | 0.07 | 0.63 |
| CD19+ | 0.71 | 0.01–6.40 | 1.99 | 0.01–8.88 | 0.04 | 0.69 | 0.25–8.77 | 0.45 | 0.07–1.52 | 0.16 | 0.62 | 0.03 |

**Notes.**

p1, ALD males vs ALD females; p2, female controls vs male controls; p3, ALD males vs male controls; p4, ALD females vs female controls.
**Table 3** Correlations of the frequencies (%) of PD-1 and PD-L1 positive lymphocytes with conventional markers of inflammation in ALD and control groups (Rho–Spearman rank correlation coefficient).

| Variable | | ALD group ( $n = 56$ ) | | | | Cotrols ( $n = 25$ ) | | | |
|---|---|---|---|---|---|---|---|---|---|
| | | CRP | NEU | NLR | WBC | CRP | NEU | NLR | WBC |
| **PD-1 positive cell subsets** | | | | | | | | | |
| CD4+ | Rho | 0.05 | 0.19 | 0.14 | 0.22 | 0,02 | 0,05 | 0,10 | −0,03 |
| | P | 0.73 | 0.17 | 0.30 | 0.10 | 0,90 | 0,80 | 0,64 | 0,87 |
| CD8+ | Rho | 0.08 | 0.11 | 0.10 | 0.14 | −0,29 | 0,18 | 0,20 | 0,07 |
| | P | 0.59 | 0.42 | 0.45 | 0.29 | 0,16 | 0,40 | 0,32 | 0,73 |
| CD19+ | Rho | 0.18 | 0.27 | 0.161 | 0.31 | 0,18 | −0,004 | 0,28 | −0,14 |
| | P | 0.20 | 0.04 | 0.24 | 0.02 | 0,38 | 0,98 | 0,18 | 0,50 |
| **PD-L1 positive cell subsets** | | | | | | | | | |
| CD4+ | Rho | 0.21 | 0.28 | 0.21 | 0.27 | −0,019 | −0,003 | −0,01 | −0,06 |
| | P | 0.14 | 0.04 | 0.12 | 0.04 | 0,93 | 0,98 | 0,94 | 0,77 |
| CD8+ | Rho | 0.19 | 0.24 | 0.23 | 0.22 | 0,05 | −0,03 | 0,06 | −0,13 |
| | P | 0.18 | 0.08 | 0.09 | 0.11 | 0,79 | 0,88 | 0,76 | 0,53 |
| CD19+ | Rho | 0.36 | 0.34 | 0.36 | 0.33 | 0,051 | −0,29 | 0,16 | −0,34 |
| | P | 0.02 | 0.009 | 0.007 | 0.01 | 0,81 | 0,15 | 0,44 | 0,09 |

**Notes.**

CRP, C-reactive protein; NEUT, neutrophils; NLR, neutrophil to lymphocyte ratio; WBC, white blood cells; p, significance level; Rho, Spearman rank correlation coefficient.

## Correlations of the frequencies of PD-1/PD-L1 positive T and B lymphocytes with conventional markers of inflammation in patients with ALD

Significant correlations of the CD19+PD-L1+ frequencies with all conventional markers of inflammation (i.e., white blood cell and neutrophil counts, C-reactive protein, and neutrophil-to-lymphocyte ratio) were found. Moreover, frequencies of CD19+PD-1+ and CD4+PD-L1+ cells revealed a positive correlation with white blood cell and neutrophil counts (see Table 3).

## Correlations of the frequencies of PD-1 and PD-L1 positive lymphocytes with liver function parameters and severity scores /MELD, CTP, mDF/

There were no correlations of any frequencies of PD-1/PD-L1 positive lymphocytes with liver enzymes (ALT, AST, AP, GTP) and its synthetic function parameters (bilirubin, albumin, urea, INR) even in relation to patients' gender.

The association between the frequencies of PD-1/PD-L1 positive lymphocytes and advanced liver dysfunction defined by MELD score >20, CTP C class, and severe alcoholic hepatitis (AH) defined by mDF >32 was also checked. The significant differences were found in the female subgroup (Tables 4, 5 and 6).

ALD females with CTP class B presented with significantly lower frequencies of PD-1 positive CD4+ T cells in comparison to CTP class A and C (Fig. S2). Also the frequencies of PD-L1 positive CD8+ cells were significantly higher in CTP class C in comparison to CTP class B (Fig. S3; Table S3).

Kasztelan-Szczerbinska et al. (2021), PeerJ, DOI 10.7717/peerj.10518

**Table 4** Comparison of the frequencies (%) of PD-1 and PD-L1 positive T and B cells in ALD patients with MELD > 20 and controls based on patients' gender (a Mann-Whitney test).

| Variable | ALD + MELD>20 | | | | p1 [a] | Controls | | | | p2 [a] | p3 [a] |
|---|---|---|---|---|---|---|---|---|---|---|---|
| | Males ( $n = 10$ ) | | Females ( $n = 3$ ) | | | Males ( $n = 16$ ) | | Females ( $n = 9$ ) | | | |
| | Median | 5-95 Percentiles | Median | 5-95 Percentiles | | Median | 5–95 Percentiles | Median | 5-95 Percentiles | | |
| **PD-1 positive cell subsets** | | | | | | | | | | | |
| CD4+ | 18.29 | 5.49–35.77 | 29.88 | 27.60–33.45 | 0.17 | 21.07 | 8.40–39.64 | 14.20 | 6.55–32.70 | 0,48 | 0.10 |
| CD8+ | 22.99 | 3.90–73.94 | 42.91 | 16.19–65.89 | 0.29 | 25.57 | 7.55–37.51 | 21.11 | 11.28–43.08 | 0,52 | 0.48 |
| CD19+ | 3.53 | 0.12–16.64 | 14.62 | 12.07–24.84 | 0.02 | 2.54 | 0.78–19.39 | 1.66 | 0.35–14.67 | 0,42 | 0.04 |
| **PD-L1 positive cell subsets** | | | | | | | | | | | |
| CD4+ | 0.74 | 0.01–2.40 | 7.18 | 4.93–11.94 | < 0.01 | 1.06 | 0.26–5.10 | 1.83 | 0.10–4.03 | 0,11 | < 0.01 |
| CD8+ | 0.45 | 0.07–1.15 | 4.71 | 2.83–8.88 | 0.01 | 0.97 | 0.13–2.45 | 0.89 | 0.15–2.88 | 0,09 | 0.02 |
| CD19+ | 0.74 | 0.01–7.48 | 12.20 | 5.06–21.01 | 0.03 | 0.69 | 0.25–8.77 | 0.45 | 0.01–1.52 | 0,67 | < 0.01 |

**Notes.**

p1, ALD males with MELD > 20 vs ALD females with MELD > 20; p2, ALD males with MELD > 20 vs male controls; p3, ALD females with MELD > 20 vs female controls.

Kasztelan-Szczerbinska et al. (2021), *PeerJ*, DOI 10.7717/peerj.10518

**Table 5  Comparison of the frequencies (%) of PD-1 and PD-L1 positive T and B cells in ALD patients with mDF > 32 and controls based on patients' gender (a Mann-Whitney test).**

| Variable | ALD + mDF>32 | | | | p1 [a] | Controls | | | | p2 [a] | p3 [a] |
|---|---|---|---|---|---|---|---|---|---|---|---|
| | Males ( n = 13) | | Females ( n = 4) | | | Males ( n = 16) | | Females ( n = 9) | | | |
| | Median | 5-95 Percentiles | Median | 5-95 Percentiles | | Median | 5–95 Percentiles | Median | 5-95 Percentiles | | |
| **PD-1 positive cell subsets** | | | | | | | | | | | |
| CD4+ | 18.88 | 4.94–35.50 | 28.74 | 20.54–33.45 | 0.20 | 21.07 | 8.40–39.64 | 14.20 | 6.55–2.70 | 0.50 | 0.07 |
| CD8+ | 24.19 | 4.30–68.71 | 33.24 | 16.19–65.89 | 0.35 | 25.57 | 7.55–37.51 | 21.11 | 11.28–3.08 | 0.56 | 0.50 |
| CD19+ | 3.21 | 0.21–15.73 | 13.34 | 4.22–24.84 | 0.03 | 2.54 | 0.78–19.39 | 1.66 | 0.35–4.67 | 0.75 | 0.02 |
| **PD-L1 positive cell subsets** | | | | | | | | | | | |
| CD4+ | 0.73 | 0.09–2.45 | 6.05 | 0.83–11.94 | 0.01 | 1.06 | 0.26–5.10 | 1.83 | 0.10–4.03 | 0,12 | 0.05 |
| CD8+ | 0.43 | 0.07–1.49 | 3.77 | 1.20–8.88 | < 0.01 | 0.97 | 0.13–2.45 | 0.89 | 0.15–2.88 | 0,10 | 0.03 |
| CD19+ | 0.54 | 0.01–7.43 | 8.63 | 1.47–21.01 | 0.02 | 0.69 | 0.25–8.77 | 0.45 | 0.01–1.52 | 0,66 | 0.01 |

**Notes.**

p1, ALD males with mDF > 32 vs ALD females with mDF > 32; p2, ALD males with mDF > 32 vs male controls; p3, ALD females with mDF> 32 vs female controls.

Kasztelan-Szczerbinska et al. (2021), *PeerJ*, DOI 10.7717/peerj.10518

**Table 6** Comparison of the frequencies (%) of PD-1/PD-L1 positive T and B lymphocytes in ALD patients with CTP class C versus controls based on patients' gender.

| Variable | ALD and CTP class C | | | | p1* | Controls | | | | p2* | p3* |
|---|---|---|---|---|---|---|---|---|---|---|---|
| | Males ($n = 13$) | | Females ($n = 4$) | | | Males ($n = 16$) | | Females ($n = 9$) | | | |
| | Median | 5–95 percentile | Median | 5–95 percentile | | Median | 5–95 percentile | Median | 5–95 percentile | | |
| **PD-1 positive cell subsets** | | | | | | | | | | | |
| CD4+ | 17.29 | 4.90–35.64 | 28.74 | 20.54–33.45 | 0.10 | 21.07 | 8.40–9.64 | 14.20 | 6.55–30.67 | 0.24 | 0.07 |
| CD8+ | 22.99 | 4.17–70.50 | 33.24 | 16.19–65.89 | 0.32 | 25.57 | 7.55–7.51 | 21.11 | 11.28–43.08 | 0.40 | 0.50 |
| CD19+ | 3.53 | 0.24–16.03 | 13.34 | 4.22–24.84 | 0.04 | 2.54 | 0.78–9.39 | 1.66 | 0.35–14.70 | 0.32 | 0.02 |
| **PD-L1 positive cell subsets** | | | | | | | | | | | |
| CD4+ | 0.74 | 0.02–2.45 | 6.05 | 0.83–11.94 | 0.02 | 1.06 | 0.26–5.10 | 1.83 | 0.10–4.03 | 0.14 | 0.0503 |
| CD8+ | 0.45 | 0.07–1.51 | 3.77 | 1.20–8.88 | 0.005 | 0.97 | 0.13–2.43 | 0.89 | 0.18–2.88 | 0.12 | 0.03 |
| CD19+ | 0.78 | 0.01–7.45 | 8.63 | 1.47–21.01 | 0.03 | 0.69 | 0.25–8.77 | 0.45 | 0.07–1.52 | 0.74 | 0.006 |

**Notes.**

Mann-Whitney test

p1, ALD males with CTP class C vs ALD females with CTP class C; p2, ALD males with CTP class C vs male controls; p3, ALD females with CTP class C vs female controls.

Moreover, ALD females with MELD >20, mDF>32 and Child class C differed significantly from ALD males with MELD>20, mDF >32 and Child class C in regard to the frequencies of PD-1 positive B and all subsets of PD-L1 positive lymphocytes (Tables 4, 5 and 6; Figs. 2, 3 and 4).

Moreover, our study revealed, that positive correlations existed between frequencies of PD-L1 positive CD8+ cells and mDF score in ALD females (Rho 0.48; $p = 0.04$) (Table 7). The ROC curve of CD8+PD-L1+cells in predicting mDF above 32 showed sensitivity 100.00 (95%CI [47.8–100.0]); specificity 69.23 (95%CI 38.6 –90.9), and an AUC 0.877 (95%CI [0.64–0.98]). The results were presented in Table S8 and Fig. S3. Applications of ROC curves include assessment of the effectiveness of continuous diagnostic parameters in distinguishing between diseased and non-diseased individuals. As reported elsewhere, AUC has a real clinical significance when its value exceeds 0.7 and AUC values between 0.8 and 0.9, as were found in our studied cohort, demonstrate excellent diagnostic accuracy (*Farcomeni & Ventura, 2012*; *Qin & Hotilovac, 2008*).

### Correlations between the frequencies of PD-1 and PD-L1 positive lymphocyte subsets and the presence of ALD complications i.e., ascites, hepatic encephalopathy, esophageal varices, hepato-renal syndrome, and survival

Correlations between the frequencies of PD-1/PD-L1 positive lymphocyte subsets and ALD complications were also checked in the studied cohort. Of ALD complications, only patients with ascites presented with significantly higher frequencies of PD-L1 positive CD19+ cells in comparison to their non-ascitic counterparts ($p = 0.03$) and the control group ($p = 0.02$) (see Table S6). No differences in the frequencies of PD-1+/PD-L1+ lymphocyte subsets were found concerning hepatic encephalopathy (HE), esophageal varices (EV), renal impairment, nor 30-day survival.

## DISCUSSION

Innate and adaptive immune responses represent a driving force in the initiation and development of ALD leading to liver fibrosis and subsequent conversion to cirrhosis with eventual further progression to liver cancer in about 2% of cirrhotics (*Orman, Odena & Bataller, 2013*). Regarding advanced liver disease, ALD is the main cause of cirrhosis worldwide, accounting for 50% of the cases (*Poznyak & Rekve, 2014*). Liver dysfunction with loss of the ability to clear immunogenic cellular residues and ethanol metabolites from the systemic circulation results in persistent stimulation of the immune system. Patients who suffer from ALD present not only with the increased titer of circulating antibodies but also with other lymphocyte-mediated responses triggered by antigens originating from oxidative stress (*Sutti, Bruzzì & Albano, 2016*). Paradoxically, these patients present with enhanced immune activity and exacerbated inflammatory responses, but are unable to cope with bacterial infections. Their immune effector cells are primed, but antibacterial functions are switched-off (*Riva & Chokshi, 2018*). The PD-1/PD-L axis represents an important co-inhibitory pathway, which modulates immune system activation and tolerance. However, overexpression of these molecules may contribute to excessive inhibitory signals and lead

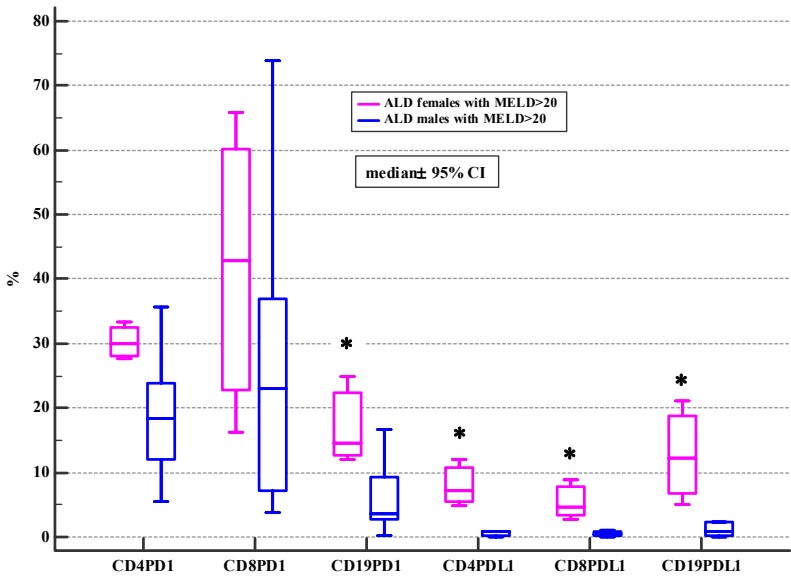

**Figure 2** Comparison of the frequencies (%) of PD-1 and PDL-1 positive T and B cells in ALD males and females with MELD > 20 (* $p < 0.05$).

to an impaired immune response. The PD-1/PD-L1 signaling has been shown to play a pivotal role in different disorders including chronic infections, autoimmune diseases, and cancers (*Riva & Chokshi, 2018*; *Pardoll, 2012*). To date, the effects of checkpoint blockade have not been fully explained in patients with ALD, the disease with no currently approved pharmacological treatment. Although many studies on immune alterations in ALD have been conducted during the last decade, the available body of evidence for the role of the PD-1/PD-L1 signaling is fairly scarce.

In general, our results revealed no differences in the frequencies of PD-1/PD-L1 positive T and B lymphocytes between ALD patients and healthy controls (Table S1). Interestingly, the analysis based on patients' gender revealed significant disparities. The studied women with ALD presented with significantly higher frequencies of PD-1 and PD-L1 positive B cells (CD19+) in comparison with female controls (Fig. 1, Table 2). Recently, the effect of PD-1 engagement in inducing B cell dysfunction has been also described in HIV infection (*Moir & Fauci, 2014*; *Boliar et al., 2012*). Furthermore, *Huang et al. (2016)*, reported that the density of CD20 positive B cells was significantly increased in the liver tissues of patients with chronic liver disease (CLD) of different etiologies compared to normal liver tissues. CLD patients with a higher hepatitis grade presented with significantly more intense B cell infiltration compared to those with a lower grade of inflammation. Enrichment of B cells in hepatic diseases may implicate this subpopulation in their pathogenesis. Since the development of collateral circulation, as a consequence of portal hypertension, facilitate the interaction of gut-derived antigens and endotoxins (that bypass the liver) with antibody-producing cells, hyperimmunoglobulinemia is frequently seen in patients with ALD (*Husby et al., 1977*; *Tomasi & Tisdale, 1964*). Alterations in the B cells number in heavy alcoholics

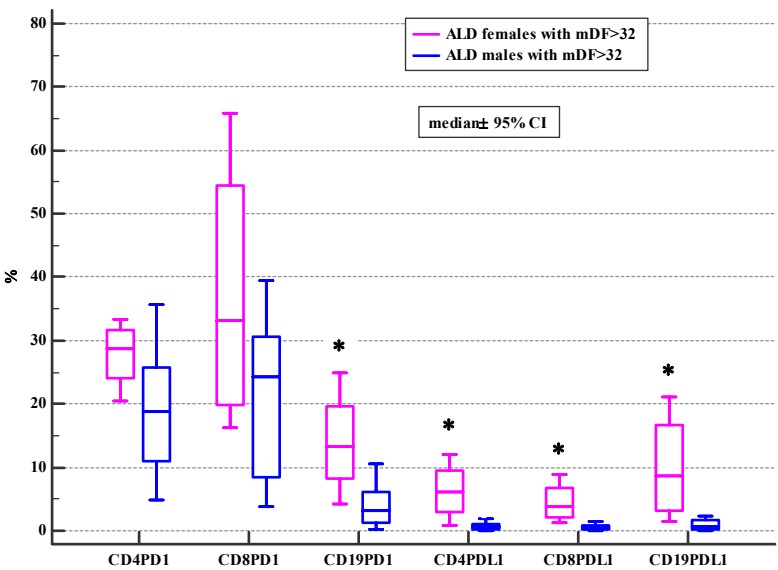

**Figure 3** Comparison of the frequencies (%) of PD-1 and PDL-1 positive T and B cells in ALD males and females with mDF > 32 (* *p* < 0.05).

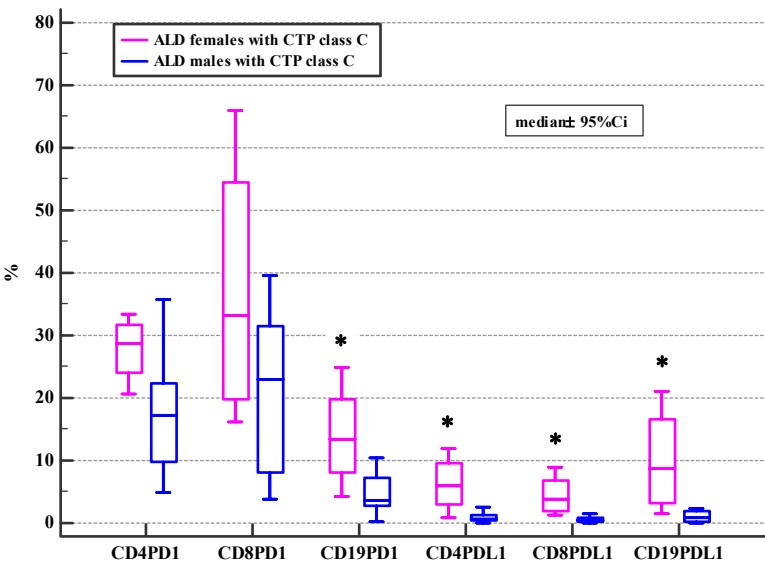

**Figure 4** Comparison of the frequencies (%) of PD-1 and PD-L1 positive T and B cells in ALD patients with CTP class C based on patients' gender (* *p* < 0.05).

and patients with ALD have been reported in previous studies (*Matos et al., 2013*; *Zhang & Meadows, 2005*). B cells through the immunoglobulin secretion may mediate both T cells-dependent and—independent immune responses (*Li et al., 2019*; *Parra, Takizawa & Sunyer, 2013*). Short-term ethanol treatment at high dose down-regulated splenic macrophages and DCs activity via enhancing B cells function as the antigen-presenting

**Table 7  Correlations of the frequencies (%) of PD-1 and PD-L1 positive lymphocytes with mDF, MELD and CTP scores in ALD females (Rho-Spearman rank correlation coefficient).**

| Variable | | mDF | MELD | CTP |
|---|---|---|---|---|
| **PD-1 positive cell subsets** | | | | |
| CD4+ | Rho | 0.03 | −0.06 | −0.06 |
| | P | 0.91 | 0.80 | 0.82 |
| CD8+ | Rho | 0.09 | 0.04 | 0.02 |
| | P | 0.73 | 0.88 | 0.93 |
| CD19+ | Rho | 0.19 | 0.18 | 0.27 |
| | P | 0.46 | 0.47 | 0.27 |
| **PD-L1 positive cell subsets** | | | | |
| CD4+ | Rho | 0.35 | 0.20 | 0.21 |
| | P | 0.16 | 0.43 | 0.40 |
| CD8+ | Rho | 0.48 | 0.29 | 0.27 |
| | P | 0.043 | 0.24 | 0.29 |
| CD19+ | Rho | 0.26 | 0.12 | 0.16 |
| | P | 0.29 | 0.63 | 0.53 |

**Notes.**
CTP, Child-Turcotte- Pugh score;  MELD,  Model for End-Stage Liver Disease;  mDF,  Maddrey's Discriminant Function;  Rho,  Spearman rank correlation coefficient.

cell, and eventually facilitating a microenvironment that leads to increased activation of CD4[+] T cells (*Andrade et al., 2009*). The relevance of B cells in patients with ALD deserves to be further clarified.

Furthermore, significant differences were found between females and males in the ALD group. They differed in terms of the frequencies of PD-1 and PD-L1 positive B cells, as well as the PD-L1 positive all T subsets. The aforementioned gender-related immune disparities occurred as a result of the disease because they were not observed in the control group (Table 2).

Our findings are consistent with recently published reports that address the role of sex hormones in modulating PD-1/PD-L1 signaling and gender-related efficacy of their blockage (*Bardhan, Anagnostou & Boussiotis, 2016*; *Wu et al., 2018*; *Capone et al., 2018*). Sex hormones influence both innate and adaptive immune responses by their impact on T and B cells, antigen presentation and cytokine secretion (*Klein & Flanagan, 2016*). It shed light on the gender-related discrepancy observed in immune disorders. Also, our results confirmed the presence of the sexual dimorphism in immune reactivity during the course of ALD due to a divergence between males and females in the frequencies of PD-1/PD-L1 positive T and B lymphocytes. Previous reports have already provided some evidence of the gender-related differences in morbidity among alcohol abusers, but underlying mechanisms are not well understood (*Chou, 1994*; *Greenfield, 2002*).

Drinking alcohol presents a health challenge for women. Even small amounts of alcohol affect women in a more severe way than men and are much more risky for them. Also, our study introduced the evidence, that females with the severe course of ALD defined by MELD>20, CTP class C or mDF>32 presented with the higher frequencies of PD-1 and PD-L1 positive B cells (CD19+) and the PD-L1 positive T cell subsets (CD4+, CD8+)

in comparison not only with female controls, but also with ALD females whose MELD was <=20 and mDF<=32, as well as with their male counterparts with MELD>20, CTP class C and mDF>32 (Tables 4, 5 and 6; Figs. 2, 3 and 4). Moreover, women with ALD classified as the CTP class C had significantly higher frequencies of PD-L1 positive CD8+ T lymphocytes in comparison to ALD females with CTP class B (Fig. S3). Also, frequencies of CD4+PD-1+ were significantly lower in females with CTP class B compared with femalres classified as CTP class A and C (Fig. S2). Our findings may suggest the impairment in the regulation of immune suppressive mechanisms in women with the severe ALD course. Further functional tests should be performed to confirm our assumption. The results are also consistent with previous reports indicating that PD-L1 is ubiquitously expressed at low levels and strongly induced by proinflammatory signals (*Liang et al., 2003*). Similar data were obtained by *Markwick et al. (2015)*, who performed the ex-vivo evaluation of immunological features of patients with acute alcoholic hepatitis (AAH) and reported hyperexpression of the PD-1 and PD-L1 proteins on T cells. Moreover, the researchers demonstrated an impaired function of T lymphocytes obtained from AAH patients what was manifested by decreased production of gamma interferon. Increased numbers of IL-10 synthesizing T cells in response to chronic endotoxin exposure were also observed. Blocking PD-1 and TIM3 reversed the aforementioned effects and restored the antimicrobial activity of both T lymphocytes and neutrophils. The findings link the PD-1 pathway to the regulation of antibacterial immunity.

The PD-1/PD-L1 expression on peripheral T and B lymphocytes was also found to be increased in other inflammatory disorders. *Jia et al. (2016)*, reported an upregulation of PD-1 expression on CD4+ and CD8+ T cells in severe septic patients and in patients with type 2 diabetes mellitus (T2DM). However, patients with severe sepsis showed higher PD-1 expression in comparison with T2DM patients.

The increased expression of PD-1 on T cells and PD-L1 on monocytes was observed in patients with septic shock in comparison with healthy controls (*Zhang et al., 2011*). However, in contrast to our study, the PD-1 expression on B cells (CD19+) and the PD-L1 expression on T (CD4+, CD8+) and B cells (CD19+) were not changed.

There are many similarities between severe ALD and sepsis in the context of the immune derangement. Patients with alcohol-associated liver failure, alike septic patients, present with an increased susceptibility to infections as a result of concurrent immune activation and immune exhaustion that seems to be related to the overexpression of negative immune checkpoints causing immunosuppression (*Riva & Chokshi, 2018*). It becomes more clear now, why several attempts to attenuate inflammatory response by the administration of antibodies directed against tumor necrosis factor-alpha have brought disappointing results in severe AH (*Naveau et al., 2004*).

The PD-1 overexpression on neutrophils in patients with systemic lupus erythematosus (SLE) correlated with the disease activity and severity. Similarly in our study, the frequencies of PD-L1 positive CD8+ correlated with mDF in ALD females (Table 7). The results of *Liu, Weng & Weng (2009)*, showed that patients with SLE had significantly increased percentages of PD-1-expressing CD3+T cells and CD19+B cells, PD-L1-expressing CD19+ B cells, and PD-L2-expressing CD14+ monocytes. ALD females in our study also presented

with significantly higher frequencies of the PD-1/PD-L1 positive B cells (CD19+) compared to female controls (Fig. 1, Table 2).

During the time of regular immune response, the upregulation of PD-1 expression occurs in response to T cell activation. Then interaction with PD-L1 and PD-L2 appears to deliver negative signals and eventually lead to the apoptosis of activated lymphocytes (*Sharpe et al., 2007*). Several reports reveal that persistent hyperexpression of the co-inhibitory molecules promotes T cell exhaustion with a progressive loss of their effector function. Exhausted T cells express multiple inhibitory receptors, that take part in the negative immune regulation. Therefore, the checkpoint inhibitors affect the host response to infection by modulating the balance between effective immune defense and immune-mediated tissue injury (*Fallon et al., 2018*).

Microorganisms may take advantage of the inhibitory signals in order to escape immune surveillance and favor chronic infection (*Keir et al., 2008*). Therefore, overexpression of PD-1/PD-L1 is a major feature of chronic infections in humans (*Golden-Mason et al., 2007*; *Day et al., 2006*; *Boni et al., 2007*). Consequently, findings in our female subgroup may also suggest that their antibacterial defense is impaired. On this basis, we are tempted to speculate that women with ALD may be more than men prone to infections, that frequently complicate the course of the disease (*Jalan et al., 2014*). The issue needs to be explored in future trials.

*Xu et al. (2014)*, demonstrated that PD-1 expression on T cells was changing along with the HBV infection progression and its variation was correlated with HBV virus load as well as liver function. We also observed differences in the frequencies of PD-1/PD-L1 positive T and B cells associated with the progression and severity of the ALD course in females (Tables 4, 5 and 6; Figs. 2, 3 and 4). Therefore, the expression patterns seem to change with the disease evolution, from the early stage of steatosis through alcoholic hepatitis to liver cirrhosis, and finally to liver cancer, and await further exploration.

Moreover, the available body of evidence indicates that the PD-1/PD-L1 pathway can impact the immune clearance of antigen-presenting cells and T lymphocytes, and promote cancer development. Therefore, it has emerged as a promising new target for cancer therapy due to a good treatment response seen in metastatic renal and lung carcinomas, as well as melanomas (*Philips & Atkins, 2015*). Whether the expansion of the PD-1/PD-L1 positive lymphocytes demonstrated in our studied women with ALD is relevant in terms of the risk of neoplasm development needs to be further clarified.

Our results are particularly noteworthy in the light of alarming findings of the recent publication authored by *Grant et al. (2017)*. The researchers reported increases in the proportion of women who drink alcohol and in high-risk drinking patterns among women, as well as the higher 12-month prevalence of alcohol use disorders among women.

Also, a recent study by *Lowe et al. (2019)*, points to gender differences in alcohol preference and indicates that inhibition of various steps in inflammasome signaling can reduce alcohol consumption in females. Therefore, targeting an inflammatory cascade in ALD seems to open new insights into the development of modern treatments focused on individually tailored therapy.

If the results of our study are confirmed, several potential prospects emerge from them.

1. Since gender-related disparities in regulatory immune mechanisms exist in ALD patients, there is a strong need to re-arrange the treatment regimens.
2. Host-directed immunomodulatory therapies aimed at re-establishing the impaired immunity create an encouraging approach to the management of patients with ALD.
3. Targeting the PD-1/PD-L1 pathway seems to be helpful in amelioration of immune-dysregulation and therefore requires further in-depth investigations in ALD.

Our findings should be interpreted with caution in the context of potential limitations. First, this was a single-center and a pilot study, therefore the overall sample size is relatively small. Second, we have not evaluated the changes of the PD-1/PD-L1 expression in the same individual over time, therefore, the impact of treatment effects cannot be assessed. Third, since the present study was not designed to predict the mortality of ALD in relation to the PD-1/PD-L1 expression, the relatively small number of non-survivors (only 5 persons) makes the evaluation likely biased. Fourth, alcohol consumption was self-reported in our study. As demonstrated by *Stockwell & Stirling (1989)*, most people are not able to accurately assess the volume and power of a drink, so the real ethanol intake may lack precision. The impact of recall bias, as well as a deliberate misreporting of alcohol consumption also cannot be ruled out.

## CONCLUSIONS

Our research adds new evidence to a limited number of studies conducted on the role of checkpoint inhibitors in alcohol abusers with ALD. The findings show that gender-related differences in the frequencies of PD-1/PD-L1 positive T and B cells in the peripheral blood exist and might potentially be related to different susceptibility to ethanol-induced liver damage in men and women. Since the upregulation of the frequencies of PD-1/PD-L1 positive lymphocytes paralleled both the severity of AH and liver dysfunction in females with ALD, the pathway seems to play a pivotal role in the disease progression. Results from our study may be further exploited to re-define therapeutic targets and create sex-tailored interventions for ALD treatment according to individual patient needs. Consequently, our findings represent an initial step in the exploration of ethanol- and sex-related alterations in immune cells reactivity modulated by the PD-1/PD-L1 pathway.

### Funding
This study was supported by the Research Grants from the Medical University of Lublin, Poland (PW369; 2015-2017). The funders had no role in study design, data collection and analysis, decision to publish, or preparation of the manuscript.

### Grant Disclosures
The following grant information was disclosed by the authors:
Medical University of Lublin, Poland: PW369.
## Competing Interests

The authors declare there are no competing interests.

## Author Contributions

- Beata Kasztelan-Szczerbinska conceived and designed the experiments, performed the experiments, analyzed the data, prepared figures and/or tables, authored or reviewed drafts of the paper, and approved the final draft.
- Katarzyna Adamczyk, Agata Surdacka, Agata Michalak and Mariusz Szczerbinski performed the experiments, prepared figures and/or tables, and approved the final draft.
- Jacek Rolinski conceived and designed the experiments, analyzed the data, authored or reviewed drafts of the paper, and approved the final draft.
- Agnieszka Bojarska-Junak analyzed the data, authored or reviewed drafts of the paper, and approved the final draft.
- Halina Cichoz-Lach conceived and designed the experiments, analyzed the data, authored or reviewed drafts of the paper, and approved the final draft.

## Human Ethics

The following information was supplied relating to ethical approvals (i.e., approving body and any reference numbers):

The study protocol conforms to the ethical guidelines of the 1975 Declaration of Helsinki (6th revision, 2008) as reflected in a priori approval by the institutional review board of Medical University of Lublin (KE-0254/141/2010).

## Data Availability

The raw measurements are available in the Supplemental Files.

## Supplemental Information

Supplemental information for this article can be found online at http://dx.doi.org/10.7717/peerj.10518#supplemental-information.

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
