# Peer review of "Gender-related disparities in the frequencies of PD-1 and PD-L1 positive peripheral blood T and B lymphocytes in patients with alcohol-related liver disease: a single center pilot study"

_PeerJ, doi:10.7717/peerj.10518_

## Round 0.1 · original submission · Major Revisions

Dear Authors, please pay special considerations to the comments dealing with the presentation of the ideas and results. In the present form, it seems a not well-designed study with a marginal contribution to the field.

Reviewer 1 ·

Basic reporting

The manuscript entitled “Gender-related disparities in the frequencies of PD-1 and PD-L1 positive peripheral blood T and B lymphocytes in patients with alcohol-related liver disease - a single center pilot study” is an original research, well written (minor correction) and well-structured article that clearly highlights differences in PD1+ and PDL1+ T and B cells in patients with ALD, particularly focusing in a gender-based analysis. Literature references can be improved in the introduction.

Line 47 …well as PD-L1+ all T cell subsets in comparison with ALD males
Line 54 Define AH
Line 73 new diagnostic and therapeutic tools and reach new conclusions should be fostered
Line 74 In comparison to huge advances in the management of viral hepatitis (vaccines and oral therapies for HBV, oral regimes for HCV), alcohol-related liver disease (ALD) management has lagged.

Line 86 … that affect the function of multiple organs and tissues, potentially causing their failure

Line 96 ….PD-1 conducts signals only when IT is cross-linked with B- or

Lines 101-113 Contextualize the known evidence of PD1 in alcohol consumption and liver disease. Examples:

https://pubmed.ncbi.nlm.nih.gov/25479137/
https://pubmed.ncbi.nlm.nih.gov/30949049/
https://pubmed.ncbi.nlm.nih.gov/29935904/

Line 114 Since a great body of evidence indicates that compared with their male counterparts, women are more susceptible to the toxic effects of ethanol in the liver for any given dose of alcohol (Please reference this statement adequately) Like the reference stated in Line 234

Line 253 positive B cells, as well as PD-L1 positive T cells (all subsets) in comparison with ALD males

Table 1. Explain what is the p significance related to
283 CTP class B (Supplementary Table S8).

Experimental design

This works present original research through a methodological design that is adequate to determine the PD-1/PDL-1 expression on peripheral T and B lymphocytes, its correlation with markers of inflammation and the severity of liver dysfunction in the course of alcohol related liver disease (ALD) and with a gender focus that has revealed interesting differences.

Major comments:

Flow cytometry data regarding IMF and not only percentage of + cells should be analyzed and shown. Why are CD3+ cells (T-cell marker) shown when no triple staining (optimal) was done. Is it not enough showing CD4+ and CD8+ cells? If this marker is of some utility then it should be addressed in the analysis and discussion.

Validity of the findings

The results presented in this work are of a high impact to understand ALD. Nonetheless, it is important to address some of the following comments regarding how results are presented and discussed. I particularly urge the authors to improve how they explain the specific consequences of CD4+ and/or CD8+ cell dysfunction in the context of the results and the disease. What should be expected if these cells have an abnormal immune response, how should this issue be addressed. In the case of B cells, serum Ig should be measured. What functional tests should be done to confirm that they have an abnormal immune response?

1.- Line 257 Analysis of mutual correlations between frequencies of PD-1/PD-L1 positive B and T lymphocyte subsets in patients with ALD.
In what table is this shown and what does this mean as it is not discussed.

2.- Line 264 Correlations of the frequencies of PD-1/PD-L1 positive T and B lymphocytes with conventional markers of inflammation in patients with ALD.
Why are CD3 and CD8+ cells for PD1 and PDL1 not shown?

3.- Line 266 Significant correlations of the CD19+PD-L1+ frequencies with all conventional markers of inflammation (i.e. white blood cell and neutrophil counts, C-reactive protein, and neutrophil-to-lymphocyte ratio) were found.
Include controls

4.- Line 284. "Moreover, ALD females with MELD >20 or mDF>32 differed significantly from ALD males with MELD>20 or mDF >32 in regard to the frequencies of PD-1 and PD-L1 positive lymphocytes (Table 5, 6, Supplementary Fig. S2, 3). "
Please describe what subsets were affected and which not

5.- I suggest merging tables for more integrated analysis, probably merging tables 3 and 4 and tables 5 and 6.

6.- “The association between the frequencies of PD-1/PD-L1 positive lymphocytes and
277 advanced liver dysfunction defined by MELD score >20 and CTP B and C class, and alcoholic hepatitis (AH) defined by mDF >32 was also checked. The significant differences were found in the female subgroup (Table 3, 4)”
No CTP B and C class are mentioned in those tables.

7.- Table 1 should show data from control subjects

8.- Why would only B cell PD-1 + lymphocytes increase in frequency? Can this differential increased expression between PD-1+ be explained?

9.- “Line 365 Our findings demonstrate the impairment in the regulation of immune suppressive mechanisms in women with the severe ALD course.”
There is no evidence of impairment, only the assumption based on markers. Functional tests should be performed.

·

Basic reporting

In the present work, it suggests that there is a differential response in T and B cells in women with ALD by measuring the percentage of cells positive for PD1 and PDL-1. It was observed that in the general population there are no differences concerning healthy controls, however, when analyzed by gender, women with ALD presented significant differences for their healthy controls.
In this work, the expression of PD1 and PDL1 is defined as a marker of damage that can be associated with future complications such as neoplasms resulting from alcohol consumption. PDL-1 is associated with the presence of ascites, in addition to hypergammaglobulinemia in nonalcoholic liver diseases. PD1 and PDL-1 promote lymphocyte depletion by giving inefficient responses with a loss of functionality and antimicrobial activity.
In the background, it is necessary to delve into the immunological implications of expressing PD1 or PDL-1 in T and B cells, in addition to explaining that the female ones have important immunological differences to the male gender in the inflammatory diseases, so in this case, it is necessary to establish the objective of the work, it is not clear if PD1 or PDL1 is a marker of damage, a target molecule for a possible delay in liver damage, or the appearance of cancer, it is important to justify the relevance of this finding.

Experimental design

It is important to give clarity to the results, the way of presenting them is confusing, it is necessary to incorporate the data of the controls in all the tables, or to make a single table with the different conditions to be able to visualize the data and compare and conclude them.

Although the results are overwhelming in the female with ADL, the possible role of increased PD1 or PDL1 is not discussed, it is only mentioned that they are involved in unbalanced inflammatory processes. There is no direct immunological evidence that these lymphocytes are dysfunctional.

Immunoglobulins are not measured to conclude the hyperglobulinemia observed in other liver diseases. Table 7 is not clear the results can be presented in a better way. It is important in the discussion to mention the relevance of PD1 and PDL1 and its possible implication in pharmacological treatment or as a predictive marker of some neoplasm.

Validity of the findings

I consider it necessary to rethink the article and give it a more focused approach to the immunological implications of the PD and PDL1 expression in B cells in females with ADL, it is not clear what the objective of measuring it is. The article does not present any functional assay of these cells. There are limitations when working with samples, however, within the clinical history of the patients, the recurrence of infections or some other indicator that the immune response is altered would be very convenient.

I recommend presenting better tables or graphs that include controls, in addition to the fact that a better description of the test used in patients to determine ALD levels is useful, it is important to mention that they consist of the main text and not the supplementary material.

Additional comments

The results are conclusive, women with ADL increase the frequency of PD1 and PDL1 on their surfaces, however, it is necessary to justify the measurement of these markers in these liver diseases, to define what is the objective of this work, it is not clear if PD1 and PDL 1 will be used as possible markers of inflammation, complications or as a target for pharmacological treatment. The tables can be summarized and it is necessary to include the controls, although there is no difference. It would be useful to include some evidence of dysfunction in the immune system since the information is poor in biochemical and cellular markers. A medical history showing a higher incidence of infections or cancer may be helpful. The relevance of PD1 or PDL1 expression remains to be discussed, both molecules have different implications on the immune response. What is the function of PD1 and what is the function of PDL1? The response in women is different from that of men in conditions of inflammation, it is important to highlight these differences and their consequences on the body. In this case, what is the importance of expression in B cells?
Also, it is necessary to detail the selection criteria, the tests used as they were only mentioned but it is not described.

---

## Round 0.2 · accepted · Accept

I have reviewed the rebuttal letter and the new version of the manuscript. All the points raised by both Reviewers were appropriately addressed, and the new version of the manuscript included the requested modifications. The only major concern that was not addressed by the authors was the functional characterization of the analyzed cell populations. I do agree with the authors' opinion, the inclusion of these data would divert the main aim of this study, and overshadow the PD-1/PD-L1 characterization. It is most suitable to include this point in a follow-up study.